# Diversity and Advantages of Culturable Endophytic Fungi from Tea (*Camellia sinensis*)

**DOI:** 10.3390/jof9121191

**Published:** 2023-12-13

**Authors:** Thanyarat Onlamun, Autchima Boonthavee, Siraprapa Brooks

**Affiliations:** 1Department of Agricultural Biotechnology (Plant), Hungarian University of Agriculture and Life Sciences, 1118 Budapest, Hungary; onlamun.thanyarat@stud.uni-mate.hu; 2School of Science, Mae Fah Luang University, Chiang Rai 57100, Thailand; 6031105041@lamduan.mfu.ac.th

**Keywords:** diversity, endophytic fungi, *Stagonosporopsis cucurbitacearum*, antifungal, plant growth promotor

## Abstract

*Sordariomycetes*, *Dothideomycetes*, and *Eurotioycetes* are three classes of endophytes that colocalize with tea (*Camellia sinensis*). Overall, the diversity indexes in this study indicated a greater abundance of fungal endophytes in roots and stems. Taking the production system into account, conventional tea plantations exhibit lower diversity compared to organic tea plantations. Notably, the influence of agrochemicals had the largest impact on the fungal endophyte communities within roots and young leaves. Despite the limited fungal diversity in conventional plantations, three fungal endophytes were isolated from tea in this culture system: *Diaporthe* sp., YI-005; *Diaporthe* sp., SI-007; and *Eurotium* sp., RI-008. These isolated endophytes exhibited high antagonistic activity (93.00–97.00% inhibition of hypha growth) against *Stagonosporopsis cucurbitacearum*, the causal agent of gummy stem blight disease. On the other hand, endophytic fungi isolated from tea in an organic system—*Pleosporales* sp., SO-006 and *Pleosporales* sp., RO-013—established the ability to produce indole-3-acetic acid (IAA; 0.65 ± 0.06 µg/mL) and assist the solubilizing phosphorus (5.17 ± 1.03 µg/mL) from the soil, respectively. This suggested that the level of diversity, whether at the tissue level or within the farming system, did not directly correlate with the discovery of beneficial fungi. More importantly, these beneficial fungi showed the potential to develop into biological agents to control the devastating diseases in the cucurbit family and the potential for use as biofertilizers with a wide range of applications in plants. Therefore, it can be concluded that there are no restrictions limiting the use of fungal endophytes solely to the plant host from which they were originally isolated.

## 1. Introduction

Endophytic fungi are naturally occurring highly diverse plant colonizers. These microorganisms inhabit the intracellular and intercellular space of plants without showing any visible symptoms [1,2]. Endophytes have a close relationship with their host plants either by maintaining a mutually beneficial relationship or by producing secondary metabolites antagonistic to pathogens [3,4,5,6,7,8]. The metabolites and volatile compounds disintegrate several barriers before colonizing plant tissue [9,10,11]. Endophytic fungi have been isolated from various plant species, including crops, invasive plants, woody perennials, mosses, ferns, and lichens. Interestingly, their greater richness has been reported in tropical and temperate climate plant species compared to others, offering various advantages to the host plants [12,13]. However, whether they maintain the same level of diversity in commercial farming is an issue to address.

Endophytes play a substantial role in plant growth and development due to their ability to synthesize phytohormones e.g., auxin and facilitate the absorption of essential nutrients such as biological nitrogen fixation, and phosphate solubilization [14,15]. Several bacterial endophytes, including *Azotobacter*, *Azospirillum*, *Acinetobacter*, *Agrobacterium*, *Arthrobacter*, *Bacillus*, *Burkholderia*, *Pseudomonas*, *Serratia*, *Streptomyces*, *Rhizobium*, *Bradyrhizobium*, *Mesorhizobium*, *Frankia*, and *Thiobacillus*, have been proven to promote growth in cereals, legumes, fruits, vegetables, herbs, and ornamentals under greenhouse and field conditions [16]. As for fungal endophytes, it has been shown that *Monilia* sp. can produce indole-3-acetic acid (IAA), siderophores, and phosphate-solubilizing activity for plant growth and development. Similarly, *Aspergillus japonicus* displayed high concentrations of salicylic acid (SA) and IAA [17].

Besides their role in plant growth promotion, microbial-mediated suppression provides a second-layer defense to their host plant against various biotic and abiotic stresses, such as disease susceptibility, drought, salinity, insects, and temperature fluctuations [18,19,20]. Endophytes have been a less exploited niche despite their enormous potential for plant protection and drug discovery. Their wide distribution in the natural ecosystem underscores their importance as biological agents used to control plant pathogenic agents. They offer plant protection by various means, such as the production of toxic compounds to pathogens, the disruption of pathogens’ cellular membranes (followed by cell death), and occupying the ecological niche of the pathogen [21,22]. In the past, some efforts were made to control various diseases in economically important crops. For instance, *Trichoderma*, *Pestalotiopsis*, *Curvularia*, *Tolypocladium*, and *Fusarium* isolated from cacao were used to control pathogenic fungi *Phytophthora palmivora* [23]. The fungi *Chaetomium globosum* suppressed the growth of a pathogenic fungi, *Setosphaeria turcica*, in maize [24].

Gummy stem blight (GSB), caused by the fungus *Stagonosporopsis cucurbitacearum*, is a limiting factor in growing areas for cucurbits such as watermelon, cantaloupe, cucumber, pumpkin, and other melons [25]. Once the pathogen lands on the plant surface, the fungus begins to infect in spite of the plant stage. Infected leaves can exhibit dark brown circular spots that start at the margins and rapidly spread to the entire leaf, with small water-soaked spots becoming visible on the fruit. This disease can cause significant production losses under favorable conditions and is difficult to manage if not detected at the early stage [26]. Commercially, the application of fungicides is a common practice to manage this disease, but this comes with significant costs, including damage to humans [27]. Moreover, the use of fungicides presents a significant challenge in their ability to completely control the disease and possibly induce fungicide resistance [28]. Therefore, the identification of an alternate solution would be very helpful.

Fungal endophytes have been isolated from various plants, including tea plants (*Camillia sinensis*). In in vivo conditions, fungal endophytes from tea leaves such as *Daldinia eschscholtzii* and *Colletotrichum gloeosporioides* have exhibited strong antifungal activity against the notorious, broad-host-range plant fungal pathogen *Sclerotinia sclerotiorum* [29]. Additionally, *Pseudocercospora kaki* and *Penicillium sclerotiorum* have demonstrated the ability to inhibit the growth of the rice blast pathogen *Magnaporthe grisea* [30]. Another strain of *Penicillium sclerotiorum*, also isolated from tea plants, has shown potential for promoting plant growth due to its ability to produce IAA, solubilize phosphate, and solubilize zinc [31]. Previous reports have highlighted the influence of tea plant varieties and tissue types on the diversity of fungal endophytes. Interestingly, endophytic diversity did not differ significantly among three varieties of *C. sinensis*, including var. Hokumei, Sayamakaori, and Yabukita [32]. However, the authors reported on the specificity of endophytes to tea varieties and tissue type have been identified as the primary factor influencing the fungal endophytes community [32]. Specifically, a low colonization rate was observed in the stem (xylem and bark) compared to both young and old leaves [32]. Despite there are several research studies on tea fungal endophytes, including their response to production practices, inion regarding the fungal endophyte community in tea roots remains limited. Therefore, the primary objective of this study was to evaluate changes in fungi diversity based on environmental factors and/or production practices, focusing on commercial farming across four tissue types: young leaves, old leaves, stem, and root.

Furthermore, capitalizing on the potential of fungal endophytes as a valuable resource for creating biocontrol agents and biofertilizers holds significant importance. Fungal endophytes are known for their ability to produce a diverse array of valuable secondary metabolites. Consequently, it is imperative to recognize that the application of fungal endophytes as biocontrol agents should not be confined exclusively to the host plant from which they were originally isolated. Hence, the secondary objective of this study was to explore the feasibility of employing fungal endophytes from tea to enhance plant growth and for future biocontrol application across a variety of plant species. Since the GBS disease has posed a significant threat to the cucurbit family. The conventional use of chemicals in this scenario directly impacts the quality of the fruit. Consequently, this paper will evaluate their potential to inhibit the pathogenic fungus *S. cucurbitacearum*, which is responsible for GSB disease. Furthermore, their capacity to synthesize IAA and facilitate phosphate absorption will be assessed to demonstrate their role in enhancing plant growth and development.

## 2. Materials and Methods

### 2.1. Identification and Isolation of Endophytes

The most popular tea variety in Chiang Rai, Thailand, is *Camillia sinensis* var. assamica. The commercial tea plantations located at Singha Park, Chiang Rai, Thailand (19.8531° N, 99.7430° E), have implemented two different cultivation systems: one strictly follows organic practices, while the other had adhere to conventional methods for 15 years. Twenty healthy tea plants (*C. sinensis* var. assamica) with uniform growth from each conventional and organic field were selected; two tissues of each young leaf (third leaf), old leaf (sixth leaf), stem, and root were collected from each plant. The samples were carefully collected in plastic bags, immediately placed in an icebox and transported to the laboratory. Afterward, the samples were washed with running water to remove soil and then subjected to sterilization as follows: leaves were submerged in 10% sodium hypochlorite for 10 min, rinsed three times with sterile water, and dried with sterile blotting paper. The surface-sterilized samples were then cut into small pieces using sterilized blades. These plant samples were placed on Potato Dextrose Agar (PDA) media and incubated at 28 °C. Hyphae of the endophytic fungi (only colonies with distinct characteristics were selected) that emerged from the segments were transferred onto new PDA media until pure cultures were obtained.

### 2.2. Characterization, Phylogenetic Analysis, and Diversity Index

Endophytic fungi were identified by using both morphological and molecular assays. The morphological characteristics of endophytic fungi were examined using an isolated culture on PDA medium. The mycelia were examined under a light microscope. For molecular analyses, genomic DNA from fungal mycelium was extracted using the CTAB (Cyltrimethyl ammonium bromide) following the protocol described in [33]. Fungal endophytes were identified by sequencing the internal transcribed regions (ITS) using a universal primer pair consisting of ITS4 (5′-TCCTCCGCTTATTGATATGC-3′) and ITS5 (5′-GGAAGTAAAAGTCGTAACAAGG-3′; [34]). The Polymerase Chain Reaction (PCR) mixture contained the following: 1×PCR buffer A, 0.5 mM MgCl_2_, 2.5 U Taq DNA polymerase, 0.25 mM dNTP, 0.5 µM of each primer, and 1 µL of genomic DNA. The PCR was performed in a DNA Engine Thermal Cycler (Eppendorf Mastercycler, Eppendorf, Germany) at 94 °C for 10 min, followed by 30 cycles at 94 °C for 45 s, 54 °C for 45 s, and 72 °C for 45 s, followed by a final extension performed at 72 °C for 10 min. The PCR products were checked for the expected size on 1% agarose gel and were sequenced at the Scientific and Technological Instrument Center, Mae Fah Luang University. The retrieved fungal endophyte sequences were performed on BLAST search to compare the sequence homology of similar nucleotide sequences of the ITS region (Appendix A). Multiple sequence alignments were prepared using MEGA v.7 [35,36]. The phylogenetic analyses were performed for maximum likelihood at 1000 replicates bootstrapping. Bootstrap values for maximum likelihood are shown at nodes. An evolutionary model for phylogenetic analyses was selected using MrModeltest v. 3.7 [37] under the Akaike Inion Criterion (AIC). The GTR + I + G model was used for Bayesian analysis. A Bayesian analysis was conducted using MrBayes v. 3.2.1 [38]. Markov chains were run for 1,000,000 generations, and trees were sampled every 100th generation (printfreq = 100), and 10,000 trees were obtained. The initial trees were discarded (20% burn-in value), and the remaining trees were used to evaluate posterior probabilities (PP) in the majority rule consensus tree. Branches with Bayesian posterior probabilities greater than 0.90 are shown in bold.

The isolation rate (IR) was calculated by dividing the number of isolated endophytic fungi with the total number of incubated plant tissues [39,40]. The colonization rate (CR) was calculated by dividing the number of plant tissues from which one or more endophytic fungi were isolated with the total number of incubated plant tissues. Relative abundance was calculated as described in [39,40]. The Shannon–Weiner Index (H′), Evenness Index (J), and Species richness index (S) were calculated using the following formulas [40,41].
H′ = −Σ(Pi × lnPi)
J = H′/In(S)
Pi = Ni/N
where N is the total number of individuals, Ni refers to the number of individuals, and S indicates the total number of species.

### 2.3. IAA Production in Endophytic Fungi

Fungal isolates were initially screened for IAA production. The isolated endophyte disc (0.5 cm) was grown on 90 mL of Czapek broth media with 10 mL filter-sterilized L-tryptophan (1000 μg/mL). The inoculated broths were incubated in an incubator shaker at 28 °C (120 rpm). After six days, 5 mL of each culture was harvested and centrifuged at 8000 rpm for 10 min. The culture filtrate was determined by adding 2 mL of Salkowski’s reagent (2% 0.5 M FeCl_3_ in 35% perchloric acid), followed by incubation at room temperature for 20 min. The optical density (OD) was measured at 540 nm using a UV-Vis spectrophotometer (ChromTech-CT 8200; Kingtech Scientific, Taipei, Taiwan). The straight-line equation was derived from the standard curve-plotted standard IAA (Sigma Aldrich, St. Louis, MO, USA) concentrations (10, 20, 50, 100 µg/mL) against OD. The calculation for IAA in the culture filter of endophytic fungi was performed by using an equation derived from the standard curve. Three replicates were performed for each experiment.

### 2.4. Determination of Phosphate Solubilization

For the qualitative estimation of phosphate solubilization, we used Pikovskaya broth media with Ca_3_(PO_4_)_2_ as a phosphate source. Actively growing fungal endophytes (0.5 cm in diameter) were placed on the center of Pikovskaya’s agar, while an un-inoculated agar plate served as control. The halo zone was measured after three and six days of incubation at 28 °C. The phosphate solubilization index (PSI) was calculated using the formula reported in [42].
$$\frac{PSI=(Colony\;diameter+Halazone\;diameter)}{Colony\;diameter}$$

### 2.5. Extraction of Fungal Crude Extract

The endophytic fungi (15 plugs per 1 flask) were incubated in 500 mL Potato Dextrose Broth (PDB) for four weeks at 28 °C. After the incubation period, cultures were filtered through two layers of sterile sheet cloth to separate the fermentation broth from the mycelium. An equal volume of ethyl acetate was added to the collected fermentation broth. The solvent was run through a separation funnel to separate compounds from ethyl acetate. The broth was extracted again with the same volume of ethyl acetate. The crude extracts from each endophyte were dried and then diluted with Dimethylsulfoxide (DMSO) to obtain a final concentration of 60 mg/mL.

### 2.6. In Vitro Assay

#### 2.6.1. Disc Diffusion Assay

The strain of fungal pathogen used, *Stagonosporopsis cucurbitacearum*, BB-001, was kindly provided by Chai Tai Co., Ltd., Chiang Mai, Thailand. The antifungal activities of the crude extracts were examined using a disc assay. Plugs of actively growing pathogenic fungi (*S. cucurbitacearum*, BB-001) were placed in the center of PDA plates. The plates were incubated at 28 °C for 72 h. Discs containing 0.6 mg of fungal crude extract, 0.5 mg of Carbendazim, and DMSO were prepared. The discs were left to air dry under a fume hood for 30 min and then placed on the surfaces of three-day-old cultures of pathogenic fungi. The plates were incubated at 28 °C for another 24 h before observing the zones of inhibition. Each crude extract was tested in triplicate, and eight discs were observed per plate.

#### 2.6.2. Poisoned Food Assay

The crude extracts that exhibited antifungal activity in the disc diffusion assays were carried over to a poisoned food assay. The PDA media (mixed with chosen crude extracts) were prepared by mixing with PDA at concentration of 0.6 mg/mL and Carbendazim at 0.5 mg/mL. DMSO was mixed with the PDA media for use as a control treatment. Plugs of pathogenic fungi were placed in the center of those PDA plates and incubated at 28 °C. The colony diameter of pathogenic fungi was measured for three days. The experiments were performed with three replications (plate) per treatment. The antifungal activities were quantified by the respective percentages of inhibition of the growth of pathogenic isolates using the following formula:$$\mathrm{The}\;\mathrm{percentage}\;\mathrm{of}\;\mathrm{inhibition}=\frac{\left(C-T\right)}{C}\times 100$$ C = average increase in mycelium growth in the control (DMSO) plate; T = average increase in mycelium growth in the treatment (fungal crude extract, Carbendazim) plate.

#### 2.6.3. IC_50_

The PDA media were mixed with crude extract and Carbendazim (control) for 5 different total concentrations (0 mg/mL, 2.5 mg/mL, 5.0 mg/mL, 7.5 mg/mL, 10.0 mg/mL) in a Petri dish plate. Five mycelium plugs of pathogenic fungi were placed onto each plate. The diameter of each plug was measured after being incubated at 28 °C for three days, and the percentage inhibition of mycelium growth was calculated. Origin Pro software 2022 (9.9) was used to calculate IC_50_.

### 2.7. In Vivo Assay

#### 2.7.1. Detach Leave Assay

The culture of pathogenic fungi was washed with 5% Tween-20 and adjusted spore suspension to 1 × 10^6^ spore/mL using a light microscope. A total of 200 µL of spore suspension was transferred into each tube; then, 8.25 µL of ddH_2_O, DMSO, Carbendazim, and crude extracts were added into each tube. The mycelium suspension was incubated at room temperature for 24 h. Healthy melon leaves were washed with running water to remove soil and sterilized with 10% sodium hypochlorite for 10 min and then rinsed with sterile distilled water three times. The mixture of 20 µL of spore suspension and the crude extract was dropped into the leaves (3 spots per leaf); each leaf was considered as one replication, and five replications per treatment were performed. The inoculate leaves were placed in a moist box for 48 h. Following a 48 h period, no symptoms had appeared on the infected leaves. Nonetheless, this duration allowed for hypha growth on the melon leaves. So, leaves were fixed and de-stained chlorophyll with an acetic acid–ethanol solution. Clear leaves were stained with 25% Coomassie Brilliant Blue. The evaluation of disease coverage on the infected leaves was performed under a light microscope.

#### 2.7.2. Greenhouse Evaluation

The experiment was set as a randomized complete block design (RCBD) with ten plots per treatment (ddH_2_O, Carbendazim, *Diaporthe* sp., YI-005, *Diaporthe* sp., SI-007, *Eurotium* sp., RI-008) and three replications. The soil (200 g) was inoculated with endophytic fungi mycelium (0.1 g). The pathogenic fungi suspensions and fungicide (Carbendazim) were prepared as mentioned above. Those solutions were sprayed directly onto three-week-old seedlings. The inoculated seedlings were then placed into a box to maintain moisture content. The disease severity was assessed at 5, 7, 9, and 11 days after inoculation. The disease severity was scored based on grades ranging from 0 to 9, with each grade, as described by [25], indicating the following: 0 = no symptoms; 1 = yellowing on leaves (suspect of disease only); 2 = moderate symptoms (<20% necrosis) on leaves only; 3 = slight symptoms (21–45% necrosis) on leaves only; 4 = severe symptoms (>45% necrosis) on leaves only; 5 = some leaves dead, no symptoms on stem; 6 = moderate symptoms (<20% necrosis) on leaves with necrosis also on petioles and stem (<3 mm long); 7 = slight symptoms (21–45% necrosis) on leaves with necrosis also on petioles and stem (3–5 mm long); 8 = severe symptoms (>45% necrosis) on leaves with necrosis also on petioles and stem (>5 mm long); and 9 = plant dead. The ratings were based on the yellowing of the leaves, with higher scores corresponding to increased necrosis on petioles and stems.

### 2.8. Statistical Analyses

All the experiments in this study were repeated independently three times. Statistical analyses of the data from each experiment were performed using an analysis of variance (ANOVA), and the mean comparison was performed using Duncan’s multiple range test (*p* < 0.05) in SPSS version 20 (IBM, Armonk, New York, NY, USA).

## 3. Results

### 3.1. Endophyte Identification and Distribution across Plant Tissues

A total of 175 endophytic fungi were isolated from four parts (young leaf, old leaf, stem, and root) of tea plants in organic and conventional tea plantations. Among them, 95 different representatives were based on morphological characteristics (colony form, mycelium color, and reverse media color; Appendix A); however, only 66 successfully maintained their culture in the laboratory. Thus, only 66 isolates were retained for the continuation of this study. Phylogenetic analysis based on ITS gene sequencing and alignment revealed that all of these endophytic fungi belong to phylum Ascomycota, with three classes (Sordariomycetes, Dothideomycetes, and Eurotiomycetes), 18 families, and 20 genera (Appendix A, Appendix A). In general, the fungal species *Diaporthe* sp., *Pseudopestalotiopsis* sp., and *Pleosporales* sp. were found in every of plant tissue (Figure 1). On the other hand, the other 17 species were identified in one or two parts of the tissue, and ten fungal species were only found in one part of plant tissue. For instance, *Eurotium* sp., which is the only species belonging to Eurotiomycetes, was found to colocalize in root tissue only (Figure 1). The diversity indexes across four tissue types are presented in Table 1. The root and stem displayed a higher level of fungal diversity, while the old leaves showed the lowest level of diversity.

### 3.2. Community Composition between Organic and Conventional System

The dominant class in this study was Sordariomycetes, with abundances of 88% in the conventional tea plantations and 65.85% in the organic tea plantations (Appendix A). The percentage of the Dothideomycetes class in the organic tea plantations (34.15%) was higher compared to that in the conventional (8%) tea plantations (Appendix A). The diversity indexes of the fungal endophytes isolated from samples in the organic fields, including Shannon’s index, Simpson’s index, species richness, evenness, and abundance, were higher than those in the conventional ones (Table 1). This indicated that there was a higher diversity of endophyte communities in the organic fields compared to the conventional ones. The 16 species identified from the organic system included *Diapothe* sp., *Pleosporales* sp., *Fusarium* sp., *Neopestalotiopsis* sp., *Nigrospora* sp., *Pseudopestalotiopsis* sp., *Alternaria* sp., *Cladosporium* sp., *Arthrinium* sp., *Clonostachys* sp., *Colletotrichum* sp., *Guignardia* sp., *Hypoxylon* sp., *Phomopsis* sp., *Roussoella* sp., and *Phaeosphaeriopsis* sp., respectively (Figure 2). Seven species that were present in the organic system were also found in the conventional system, namely *Diapothe* sp., *Neopestalotiopsis* sp., *Pseudopestalotiopsis* sp., *Clonostachys* sp., *Fusarium* sp., *Pleosporales* sp., and *Roussoella* sp., while the other three species, *Chaetomium* sp., *Ovatospora* sp., *Pestalotiopsis* sp., were only present in the conventional system (Figure 2).

Similar to the diversity indexes, the colonization rate (CR) and isolation rate (IR) of the samples from the organic system were significantly higher than those of the samples from the conventional system (*p* = 0.00081, *p* = 0.0037; Figure 3). Interestingly, a number of species obtained from the root and young leaves samples from the conventional system exhibited a decrease of at least 50% in both CR and IR when compared to the samples from the organic system (Figure 3).

### 3.3. IAA Production and Determination of Phosphate Solubilization

All 66 strains of the fungal endophytes were also evaluated based on their ability in IAA production and phosphate solubilization. The derived equation (y = 0.0292x + 0.137) was used to quantify IAA in the culture filter of endophyte fungi. Six fungal isolates—*Pleosporales* sp., SO-006; *Cladosporium* sp., SO-022; *Pleosporales* sp., RO-013; *Phaeosphaeriopsis* sp., YO-002; *Phaeosphaeriopsis* sp., YO-008; and *Cladosporium* sp., YO-020—showed the potential to possess plant growth-promoting properties (Table 2). After six days of incubation, the *Pleosporales* sp. strain SO-006 significantly (*p* < 0.05) produced the highest concentration of IAA (0.65 ± 0.06 µg/mL), while the rest of the isolates produced IAA between 0.15 and 0.27 µg/mL. The *Pleosporales* sp. strain SO-006 displayed the highest value of phosphate solubilization (3.60 ± 0.41) after incubation for three days; however, no significant differences in this regard among the six aforementioned strains were observed after incubation for three days (Table 2). One significant difference (at 5% level) was observed after six days of incubation, with the *Pleosporales* sp. RO-013 showing the highest phosphate solubilization value (5.17 ± 1.03) among all strains.

### 3.4. Antifungal Activity of the Fungal Endophytes

Disc diffusion and poisoned food assays were conducted to evaluate the antifungal activities of 66 strains of fungal endophytes. Ten fungal endophyte extracts (10.5%) showed antifungal properties, with inhibition zones ranging from 0.23 to 0.52 cM, while the inhibition zone of the recommended fungicide, Carbendazim, was 0.43 ± 0.05 cM (Table 3).

Three fungal endophytes, namely *Diaporthe* sp., YI-005; *Diaporthe* sp., SI-007; and *Eurotium* sp., RI-008, showed consistently high antifungal activity in both assays. *Diaporthe* sp., YI-005 showed the highest antifungal activity against *S. cucurbitacearum*, BB-001, with an inhibition zone of 0.52 ± 0.07 cM and percentage growth inhibition of 88.54 ± 2.27% in the disc diffusion assay and poisoned food assay, respectively (Table 3; Figure 4). *Diaporthe* sp., SI-007 (inhibition zone of 0.33 ± 0.05 cM) and *Eurotium* sp., RI-008 (inhibition zone of 0.32 cM) also showed inhibitory activity against *S. cucurbitacearum*, BB-001 in the poisoned food assay, with mycelium growth inhibition values of 88.16 ± 1.10% and 85.89 ± 2.56%, respectively (Table 3; Figure 4). The IC_50_ values also demonstrated potent antagonistic activity against *S. cucurbitacearum*, BB-001 compared to the recommended fungicide, Carbendazim. Interestingly, two endophytic fungi exhibited lower IC_50_ values compared to the recommended fungicide (Figure 5). The IC_50_ value of Carbendazim was 3.93, while the IC_50_ value of the crude extract of three fungal endophytes—*Diaporthe* sp., YI-005; *Eurotium* sp. strain RI-008; and *Diaporthe* sp., SI-007—were 1.40, 1.75, and 5.86, respectively.

The in vivo assay was carried out using three fungal endophytes. Melon leaves inoculated with mixed mycelium suspension with sterile water and DMSO were covered with hypha of *S. cucurbitacearum*, BB-001, while Carbendazim inhibited hypha growth on melon leaves completely (100% inhibition). On the other hand, fungal crude extracts from *Diaporthe* sp., YI-005; *Diaporthe* sp., SI-007; and *Eurotium* sp., RI-008 showed high antagonistic activity, with values for the inhibition of hypha growth on melon leaves of 93.00%, 97.00%, and 95.00%, respectively.

Thus, those three fungal endophyte strains were subsequently evaluated for their ability to control gummy stem blight disease in melon caused by *S. cucurbitacearum* strain BB-001 under greenhouse conditions. The soil was inoculated with endophytic fungi mycelium at a concentration of 5%. The results were collected at 5, 7, 9, and 11 days after inoculation. In the untreated group, melon presented severe symptoms at nine and eleven days after inoculation (DAI). Interestingly, the three treatments and the recommended fungicide, Carbendazim, had almost the same efficiency for controlling *S. cucurbitacearum* strain BB-001 (Table 4; Figure 6).

## 4. Discussion

This study compared the diversity of culturable endophytic fungi on PDA media from commercial organic and conventional tea plantations. While 95 isolates were initially identified based on their morphology, only 66 isolates successfully maintained their cultures in the laboratory. Several factors can impact the ability to cultivate microorganisms under laboratory conditions, including a lack of understanding of their nutritional needs and the challenging nature of microorganisms, particularly when derived from environmental samples [43]. Despite PDA experiencing limitations in maintaining all strains of fungal endophytes in the laboratory, the 66 isolates featured in this study should be sufficient to represent the diversity of endophytic fungi in tea plants. These isolates belong to the phylum Ascomycota, with a predominant presence of two classes Sordariomycetes and Dothideomycetes. The Phylum Ascomycota, particularly the Sordariomycetes and Dothideomycetes classes, is widely recognized as the most common representative of endophytic fungi communities in tea [32] and all types of plants, such as Mangrove (*Rhizohora mucronate* [12]), coffee trees (*Coffea arabica* L. and *Coffea canephora* L. [44]), Vinca rosea (*Catharanthus riseos* [45]), Orchid (*Guarianthe skinneri* [46]), and beach vitex (*Vitex rotundifolia* [47]).

The organic system is a sustainable practice characterized by maintaining and protecting natural resources by abolishing agrochemicals which commonly result in a greater richness, evenness, and diversity of fungal endophytes. The authors of [48] found that the abundance and diversity of endophytic microbial species in corn, melon, pepper, and tomato plants were significantly higher in the organic system. Similarly, the authors of [49] reported a higher abundance and species diversity for fungal endophytes in grapevines grown in organic fields compared to conventional fields. Moreover, the authors of [50] isolated endophytic fungi from the leaves, flowers, and fruit of healthy apple trees (*Malus domestica*, BORKH) growing under conventional, integrated, and organic systems. They reported that the endophytic fungal abundance from the orchards under organic cultivation was significantly higher than that in the integrated and conventional cultivation systems. In this study, a difference in the number of isolated fungi was observed between the two systems. Notably, despite the experiment being conducted in a commercial setting, this study’s results are consistent with previous reports indicating higher diversity indexes in organic tea plantations. The use of agrochemicals in conventional systems can affect the plant life cycle and metabolism in crop plants, as well as the diversity of fungal endophytes. Thus, the reductions in the diversity, richness, and evenness of endophytic species presented in conventional farming systems is directly associated with the chemicals used in the systems.

The impact of chemical disturbance on endophytic fungal populations may differ across tissue types. One study [51] found that fungicides can disturb the populations of endophytic fungi within the stems of tea plants while having no effect on the community of fungal endophytes isolated from tea leaves. On the contrary, our investigation revealed that root and young leaves were the most affected. This observation might be attributed to the extensive and prolonged use of pesticides in conventional tea plantations for over 15 years. Consequently, it is conceivable that exposure to chemicals has disrupted the balance within the endophytic community of conventionally grown tea, including fungal endophytes in tea leaves. Furthermore, the heavy pesticide application may have led to the destruction of many prominent members of the soil community, allowing other typically suppressed species to become more abundant [15,28,51] and subsequently influencing the community of fungal endophytes in the roots of tea plants.

The diversity of fungal endophytes depends on several factors, including plant species, growth stages, soil types, environmental conditions, farming practices, geographical location, etc. Similarly, the fungal endophyte community in tea plants is also shaped by these factors. In [32], *Colletotrichum camelliae*, *Phyllosticta capitalensis*, and a *Pleosrales* sp. were reported as common endophytes found in *C. sinensis* var Yabukita cultivated in both conventional and organic plots in Japan. While the authors of [52] reported the abundances of *Collectotricum* sp. and *Pestalotiosis* sp. in *C. sinensis* var BXZ and MF from China. Our study, conducted in Thailand, revealed a higher number of *Diaporthe* sp. and *Neopestalotiopsis* in *C. sinensis* var. assamica. In this study, *Diaporthe* was the most common genus that was present in both systems of commercial tea plantations, as it represented 36% of the endophyte isolates from the conventional field. On the other hand, this genus represented only 14.63% of the isolates from the organic system. *Diaporthe* (incl. its *Phomopsis* state) has been reported as one of the most frequently encountered genera of endophytic fungi in several plant hosts [53,54,55,56,57]. Moreover, this genus has also frequently been recognized as a producer of bioactive compounds that might benefit the plant and provide biocontrol activity with antibiotic or anticancer activity [58,59,60]. Interestingly, two out of three endophytic fungi exhibited very high antifungal activity against *S. cucurbitaceous* belonging to *Diaporthe* (*Diaporthe* sp., YI-005, and *Diaporthe* sp., SI-007).

Out of the 66 fungal endophytes examined in this study, 6 fungal endophyte isolates from the organic system exhibited the capability to enhance plant growth. In particular, *Pleosporales* sp., SO-006 and *Pleosporales* sp., RO-013, hold promise as potential candidates for development as biofertilizers. This supports previous findings indicating that fungal endophytes isolated from organic agriculture often show remarkable chemical diversity with respect to their secondary metabolites, with positive effects on plant growth and resistance to biotic and abiotic stresses [61]. However, our study demonstrated that despite the low diversity of endophytic fungi in tea that is cultivated in conventional systems, this low diversity did not affect the values of those fungal endophytes, as shown in ten fungal endophytes that displayed antifungal activity against *S. cucurbitacearum* strain BB-001 isolated from tea plants grown in the conventional system. The authors of [20] suggested that the fungal endophytes in their study assisted medical plants under abiotic stress conditions by producing and accumulating some highly coveted secondary metabolites. Since, secondary metabolite production in fungi might depend on their host species and on environmental conditions [9,15,44], there is a possibility that an imbalance in the conventional system affects the secondary metabolite production of fungal endophytes. All the fungal endophytes that demonstrated antifungal activity were obtained from host plants in the conventional system. The production of valuable biologically active secondary metabolites offers promising and fascinating opportunities for the development of important and effective biocontrol agents. However, further research is needed to explore the application of *Diaporthe* sp., YI-005; *Diaporthe* sp. strain SI-007; and *Eurotium* sp., RI-008. Nonetheless, these three endophytic fungi displayed antifungal activity similar to or higher than the recommended fungicide, Mancozeb, which demonstrates the high potential of these endophytes to be developed into commercial biological agents to control pathogens that cause gummy stem blight disease in the future.

In conclusion, despite the ecological imbalance affecting the diversity of endophytic fungi, it did not limit the capability of fungal endophytes in terms of application in agriculture, especially in the context of biotic stress as well as enhancing plant growth. More importantly, this study serves as evidence that endophytic fungi can provide benefits to a wide range of plants and that there is no need to be solely restricted to the host from which they were originally isolated.

## Figures and Tables

**Figure 1 jof-09-01191-f001:**
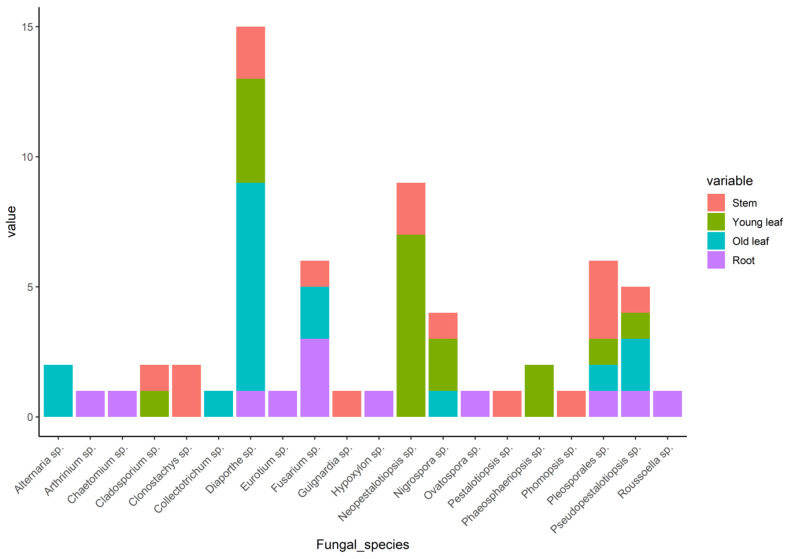
Distribution of the fungal isolates across different plant tissues.

**Figure 2 jof-09-01191-f002:**
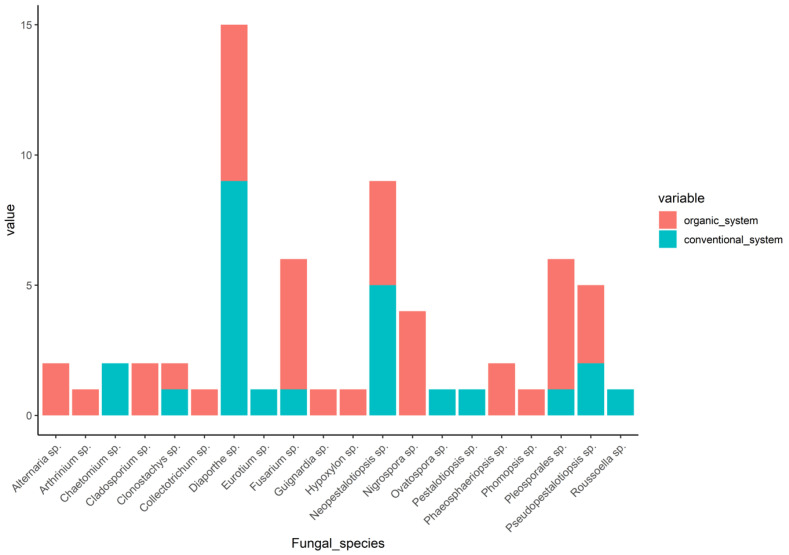
Distribution of the endophytic fungal isolated from tea plants in the organic and conventional system.

**Figure 3 jof-09-01191-f003:**
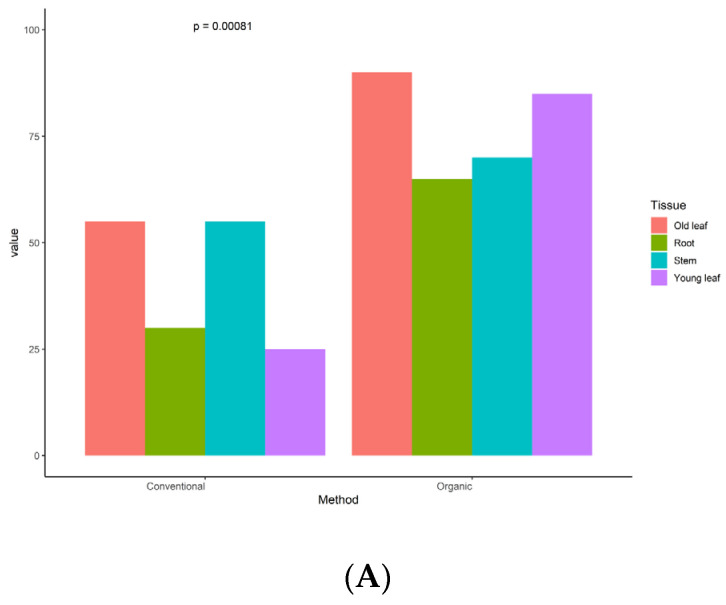
Colonization rate (CR) (**A**) and isolation rate (IR) (**B**) of the endophytic fungi isolated from the organic and conventional tea systems.

**Figure 4 jof-09-01191-f004:**
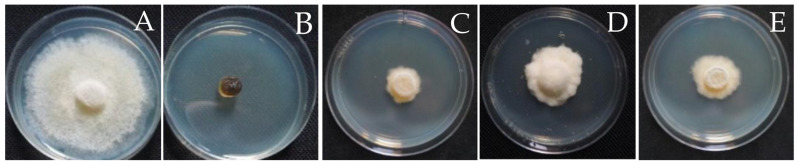
Poison food assay of (**A**) negative control-ddH_2_O; (**B**) positive control (Carbendazim); (**C**) fungal crude extracts of *Diaporthe* sp., YI-005; (**D**) fungal crude extracts of *Diaporthe* sp., SI-007; (**E**) fungal crude extracts of *Eurotium* sp., RI-008.

**Figure 5 jof-09-01191-f005:**
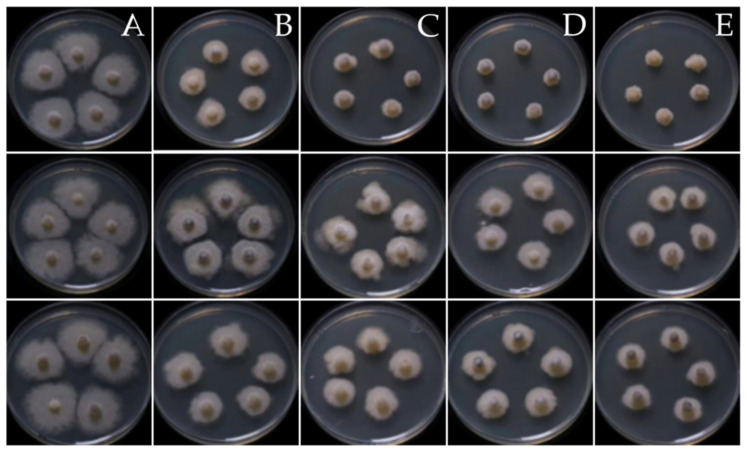
Petri dish plates to determine the IC_50_ values of fungal crude extracts using five different concentrations of crude extract: (**A**) 0 mg/mL; (**B**) 2.5 mg/mL; (**C**) 5.0 mg/mL; (**D**) 7.5 mg/mL; (**E**) 10.0 mg/mL of crude extract from *Diaporthe* sp., YI-005 (**top**); *Eurotium* sp., RI-008 (**middle**); and *Diaporthe* sp., SI-007 (**bottom**).

**Figure 6 jof-09-01191-f006:**
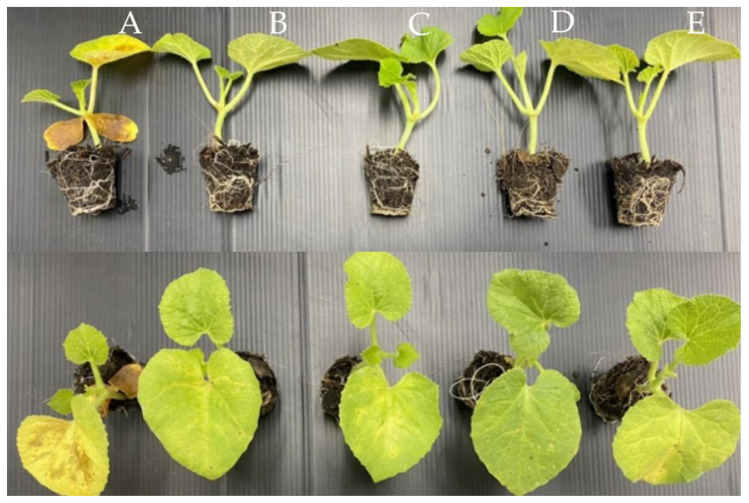
Melons after nine days of inoculation with *S. cucurbitacearum* strain BB-001: (**A**) ddH_2_O; (**B**) Carbendazim; (**C**) *Diaporthe* sp., SI-007; (**D**) *Diaporthe* sp., YI-005; and (**E**) *Eurotium* sp., RI-008.

**Table 1 jof-09-01191-t001:** Endophyte diversity indices and their estimates based on plant tissue and cultivation system.

	Species Richness	Evenness	Abundance	Shannon’s (H)	Simpson’s (λ)
Plant tissue					
Young leaves	7	0.66	15	1.78	0.87
Old leaves	7	0.56	17	1.61	0.77
Stem	11	0.72	20	2.18	0.91
Root	10	0.84	14	2.21	0.95
Cultivation system					
Organic system	16	0.83	41	2.54	1.07
Conventional system	11	0.92	25	1.99	0.84

**Table 2 jof-09-01191-t002:** IAA production and phosphate solubilization of endophytic fungi.

Code	IAA Production (µg/mL) *	Phosphate (µg/mL)	
		3 Days	6 Days *
*Pleosporales* sp., SO-006	0.65 ± 0.06 a	3.60 ± 0.33	3.84 ± 0.32 b
*Cladosporium* sp., SO-022	0.15 ± 0.03 c	3.53 ± 0.33	3.35 ± 0.45 b
*Pleosporales* sp., RO-013	0.17 ± 0.03 c	2.70 ± 0.81	5.17 ± 1.03 a
*Phaeosphaeriopsis* sp., YO-002	0.16 ± 0.05 c	3.43 ± 0.72	3.76 ± 0.85 b
*Phaeosphaeriopsis* sp., YO-008	0.27 ± 0.04 b	3.10 ± 0.38	3.88 ± 0.77 b
*Cladosporium* sp., YO-020	0.19 ± 0.04 c	2.89 ± 0.54	4.03 ± 0.41 ab

* Means with the same letter are not significantly different at 5% level.

**Table 3 jof-09-01191-t003:** Fungal extracts’ antifungal activities against *S. cucurbitacearum*, BB-001 (evaluated by conducting disc diffusion and poisoned food assays).

Fungal Crude Extract/Fungicide	Disc Diffusion Assay [Inhibition Zone (cM)] *	Poisoned Food Assay (% Mycelium Inhibition) **
Carbendazim	0.43 ± 0.05 ab	94.66 ± 2.34 a
*Diaporthe* sp., YI-002	0.27 ± 0.05 c	65.96 ± 6.57 d
*Diaporthe* sp., YI-005	0.52 ± 0.07 a	88.54 ± 2.27 ab
*Pleosporales* sp., YI-008	0.32 ± 0.07 bc	73.28 ± 2.33 c
*Neopestalotiopsis* sp., YI-102	0.41 ± 0.08 ab	41.99 ± 5.90 f
*Pseudopestalotiopsis* sp., OI-018	0.33 ± 0.05 bc	49.64 ± 0.86 e
*Pseudopestalotiopsis* sp., OI-019	0.23 ± 0.05 c	59.21 ± 4.95 d
*Neopestalotiosis* sp., SI-002	0.42 ± 0.07 ab	62.20 ± 5.56 d
*Diaporthe* sp., SI-007	0.33 ± 0.05 bc	88.16 ± 1.10 ab
*Eurotium* sp., RI-008	0.32 ± 0.07 bc	85.89 ± 2.56 b
*Ovatospora* sp., RI-012	0.33 ± 0.05 bc	35.02 ± 5.67 f

* Means with the same letter are not significantly different at 5% level. ** Means with the same letter are not significantly different at 10% level.

**Table 4 jof-09-01191-t004:** Effects of soil mixed with three different endophytic fungi on melon disease severity after inoculation with *S. cucurbitacearum* strain BB-001 under greenhouse conditions.

Treatment	Melon Disease Severity (%)			
	5DAI *	7DAI *	9DAI *	11DAI *
Untreated	31.67 ± 3.32 a	48.32 ± 13.43 a	59.65 ± 1.39 a	70.10 ± 3.00 a
Carbendazim	10.58 ± 0.91 b	30.77 ± 8.85 b	34.08 ± 10.99 b	42.16 ± 18.08 b
*Diaporthe* sp., YI-005	13.27 ± 4.37 b	24.77 ± 9.63 b	28.81 ± 7.88 b	38.18 ± 1.59 b
*Diaporthe* sp., SI-007	13.89 ± 4.81 b	28.35 ± 7.78 b	33.55 ± 3.42 b	43.74 ± 11.69 b
*Eurotium* sp., RI-008	11.73 ± 2.58 b	27.31 ± 6.26 b	37.13 ± 2.12 b	42.16 ± 2.86 b

* Means with the same letter are not significantly different at 5% level.

## Data Availability

https://www.ncbi.nlm.nih.gov/genbank/ (accessed on 27 November 2023).

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
