# Peer review of "Diversity and Advantages of Culturable Endophytic Fungi from Tea (*Camellia sinensis*)"

_jof, 2023, doi:10.3390/jof9121191_

Round 1
Reviewer 1 Report
Comments and Suggestions for Authors
The present work explores the diversity of endophytic fungi in two production systems of tea (Camellia sinensis) plants. I hope the following observations are useful:
The title of the work must include the crop being worked with, Tea (Camellia sinensis).
In the introduction, mentions the benefits of endophytic fungi, but there are endophytic fungi that produce compounds that can be harmful to humans. It is suggested to add information about the compounds that may harm the consumer.
Regarding the isolation methodology in PDA medium, it is possible that many other endophytic fungi could not be recovered since it is known that not all microorganisms grow in conventional culture media. A culture medium could have been made from tea plants.
In the figures of supplementary material, he considered that each image should include the name of the species or genus that was identified. This is so that it is important to show the images.
I consider that phylogenetic trees of the analyzed isolates must be shown and the ITS sequences must be available in databases. Which will give a lot of solidity to the work.
Author Response
- The title of the work must include the crop being worked with, Tea (Camellia sinensis).
Ans. Add in page 1
- In the introduction, mentions the benefits of endophytic fungi, but there are endophytic fungi that produce compounds that can be harmful to humans. It is suggested to add information about the compounds that may harm the consumer.
Ans. Since we are focusing on the benefits of endophytic fungi. I think it is important to present the benefic compound, we discuss the toxicity of the prototype from biocontrol in discussion part
- Regarding the isolation methodology in PDA medium, it is possible that many other endophytic fungi could not be recovered since it is known that not all microorganisms grow in conventional culture media. A culture medium could have been made from tea plants.
Ans. Thank you for your valuable comment, but unfortunately, we can’t make any changes.
- In the figures of supplementary material, he considered that each image should include the name of the species or genus that was identified. This is so that it is important to show the images.
Ans. We decided to delete supplementary figure 2 as recommended by another reviewer.
- I consider that phylogenetic trees of the analyzed isolates must be shown and the ITS sequences must be available in databases. Which will give a lot of solidity to the work.
Ans. All sequences are available in the GenBank database, and it shown in Supplementary Table 2
Reviewer 2 Report
Comments and Suggestions for Authors
In this MS, diversity of culturable endophytic fungal in organic and conventional tea production systems were studied and their role in biocontrol and enhancing plant growth were evaluated. The results are rational and useful. There are some details need to be added.
1. The title: tea should be added, …tea production system
2. How many tea trees are select as samples? How many tissues were used?
3. Please give the phylogenetic trees for the three strains of isolated fungi which based on DNA ITS sequence
Author Response
- The title: tea should be added, …tea production system
Ans. Add in page 1
- How many tea trees are select as samples? How many tissues were used?
Ans. Already in Page 4 line 124-126
Twenty healthy tea plants (C. sinensis var. assamica) with uniform growth from each conventional and organic field were selected; two tissues of each young leaf (third leaf), old leaf (sixth leaf), stem, and root were collected from each plant.
- Please give the phylogenetic trees for the three strains of isolated fungi which based on DNA ITS sequence
Ans. All phylogenetic trees present in Supplementary Figure 1
Round 2
Reviewer 1 Report
Comments and Suggestions for Authors
Dear authors, it is important to include figure captions and table captions in the supplementary material. It is striking to me that the accession numbers for the ITS sequences of the 66 isolates are not found in the databases and accession numbers are not provided.
All the access numbers they put in supplementary materials do not correspond to their 66 isolates. Can you tell why?
Author Response
Dear reviewer,
We deeply appreciate the constructive comments provided by you; please see the response to your comment
1. it is important to include figure captions and table captions in the supplementary material.
Answer. Figure captions and table captions were added to the supplementary file.
2. All the access numbers they put in supplementary materials do not correspond to their 66 isolates.
Ans. We provided 2 accession numbers in Supplementary Table 2. The 3rd column is the genebank accession that is similar to our sequence and we put our accession number of 66 isolated that we submit in GenBank in column 6. We highlight those columns with yellow labels.
Best regards,
Siraprapa
Round 3
Reviewer 1 Report
Comments and Suggestions for Authors
The suggested changes have been made. There are no further changes suggested.